# Insights into microphysical and optical properties of typical mineral dust within urban snowpack via wet/dry deposition in Changchun, Northeastern China

Tenglong Shi,[1,2,3] Jiayao Wang,[1,3] Daizhou Zhang,[4] Jiecan Cui,[5] Zihang Wang,[2] Yue Zhou,[2] Wei Pu,[2] Yang Bai,[1,3] Zhigang Han,[1,3] Meng Liu,[6] Yanbiao Liu[6], Hongbin Xie,[6] Minghui Yang,[6] Ying Li[7], Meng Gao[8] and Xin Wang[*,2,6]

[1] State Key Laboratory of Spatial Datum, College of Remote Sensing and Geoinformatics Engineering, Faculty of Geographical Science and Engineering, Henan University, Zhengzhou, China, 450046

[2] Key Laboratory for Semi-Arid Climate Change of the Ministry of Education, College of Atmospheric Sciences, Lanzhou University, Lanzhou 730000, China

[3] Henan Industrial Technology Academy of Spatiotemporal Big Data (Henan University), Zhengzhou, 450046 China

[4] Faculty of Environmental and Symbiotic Sciences, Prefectural University of Kumamoto, Kumamoto 862-8502, Japan

[5] Zhejiang Development & Planning Institute, Hangzhou 310030, China

[6] Key Laboratory of Industrial Ecology and Environmental Engineering (Ministry of Education, China), School of Environmental Science and Technology, Dalian University of Technology, Dalian 116024, China

[7] Key Laboratory of Atmospheric Environment and Extreme Meteorology, Institute of Atmospheric Physics, Chinese Academy of Sciences, Beijing, China

[8] Department of Geography, Hong Kong Baptist University, Hong Kong, China

* Corresponding author: Xin Wang ([wxin@lzu.edu.cn](wxin@lzu.edu.cn)).

**Abstract.** This study presents the first compositional analysis of dust in snowpack from

a typical Chinese industrial city, utilizing computer-controlled scanning electron

microscope combined with K-means cluster analysis and manual experience. The dust

is predominantly composed of kaolinite-like (36%), chlorite-like (19%), quartz-like

(15%), illite-like (14%), hematite-like (5%), and clay-minerals-like (4%), with minor

contributions from other components. It was also found that the size distribution and

aspect ratio of the dust did not undergo significant changes during dry and wet

deposition, but they exhibited great variability among the different mineral composition

groups. Subsequently, these observed microphysical parameters were used to constrain

the optical absorption of dust, and the results showed that under low (high) snow grain

size scenarios, the albedo reductions caused by dust concentrations of 1, 10, and 100

13    ppm in snow were 0.007 (0.022), 0.028 (0.084), and 0.099 (0.257), respectively. These

results emphasize the importance of dust composition and size distribution

characteristics in constraining snowpack light absorption and radiation processes.

**1 Introduction**

Snow constitutes a crucial component of the terrestrial cryosphere, covering approximately 40% of the global land area, with a maximum extent of around 45 million square kilometers (Hall et al., 1995; Lemke et al., 2007). It is predominantly found in polar and high-latitude regions, as well as mountainous areas at mid-to-low latitudes, exhibiting significant temporal and spatial variability due to seasonal changes (Tan et al., 2019; Thackeray et al., 2016; Zhu et al., 2021). Current research indicates that light-absorbing aerosols in the atmosphere (e.g. black carbon, brown carbon, and dust) are eventually deposited on various surfaces, including snow or glaciers through atmospheric diffusion, transport, and dry/wet deposition processes (Doherty et al., 2010; Gilardoni et al., 2022; Kuchiki et al., 2015). This alters the single optical properties of the snowfield, enhances the absorption of solar radiant energy, and reduces the albedo of the snow and ice surface, thereby accelerating snowmelt and altering the water cycle, and exerting a nuanced yet pivotal role in regional climate dynamics (Hadley and Kirchstetter, 2012; Hansen and Nazarenko, 2004; Kang et al., 2020; Skiles et al., 2018). Hence, it emerges as a critical determinant impacting both regional and global climate change.

Extensive observational evidences highlighted significant reductions in the extent and duration of snow cover across the Northern Hemisphere, particularly notable in high-latitude and mountainous regions due to global warming (Bormann et al., 2018; Derksen and Brown, 2012; Mote et al., 2018; Pulliainen et al., 2020; Zeng et al., 2018). Currently, the duration of Northern Hemisphere snow cover is decreasing by

approximately 5-6 days per decade (Dye, 2002), with Arctic June snow cover diminishing at a rate of 13.6% per decade (Derksen and Brown, 2012; Derksen et al., 2017). Regions like the western Tibetan Plateau and Australia have experienced snow cover retreat rates ranging from 11% to 30% per decade (Bormann et al., 2012; Immerzeel et al., 2009), while the onset of snowmelt in the western United States has advanced by 6-26 days since the mid-1970s (Hall et al., 2015). Dust, a prevalent aerosol type in the Earth-atmosphere system, has garnered significant scientific attention due to its role in accelerating ice and snow melt (Bryant et al., 2013; Dong et al., 2020; Kaspari et al., 2015; Painter et al., 2012). Réveillet et al. (2022) reported an 8-12 day earlier average snowmelt in the French Alps and the Pyrenees due to dust presence during 1979-2018. Zhang et al. (2018) found that dust reduced snow albedo in the southern Tibetan Plateau by approximately $0.06 \pm 0.004$, equivalent to 30% of the albedo reduction caused by black carbon. Sarangi et al. (2020) further demonstrated dust's primary contribution to snow darkening above 4000 m altitude in the Tibetan Plateau, surpassing that of black carbon in influencing regional ice and snow melt. Whereas Xing et al. (2024) and Winton et al. (2024) also highlighted the remarkable contribution of dust events to the snow darkening of the Asian High Mountains and the Southern Alps, respectively. Moreover, Hao et al. (2023) projected a decrease in black carbon deposition on ice and snow under future emission scenarios, and anticipated that heightened dust emissions and deposition fluxes driven by climate change-induced land use changes (Neff et al., 2008), frequent wildfires (Yu and Ginoux, 2022), and increased drought (Huang et al., 2016). Consequently, dust's impact on ice and snow melt is

expected to intensify markedly.

Previous studies have focused on investigating the concentration of dust in snow and its related radiative effects, neglecting the impact of the microphysical properties of dust on its optical absorption (Bryant et al., 2013; Reynolds et al., 2020; Xie et al., 2018). In fact, the physical and chemical properties of mineral dust aerosols, including their particle size distribution (PSD), composition, mixing state, and shape, determine their optical properties (Chou et al., 2008; Colarco et al., 2014; Fountoulakis et al., 2024; Haapanala et al., 2012; Shi et al., 2022b). Dong et al. (2020) compared the volume-size distribution of dust deposition in ice and snow in western China and the Arctic, finding significant differences in the median particle size of dust, and showing that the particle size decreases with altitude in various remote regions except for the remote Arctic and Antarctic regions. Wang et al. (2023) used intelligent scanning electron microscopy to obtain typical PSD of dust in snow in Changchun. Additionally, related dust studies in the atmosphere have confirmed the complex variability of dust mineral composition. For example, in the case of dust aerosols from the Sahara Desert collected in Izana, Spain, in the summer of 2005, it was found that they were mainly composed of silicates (64%) and sulfates (14%), with small amounts of carbonaceous materials (9%), quartz (6%), calcium-rich particles (5%), hematite (1%), and soot (1%) (Kandler et al., 2007). In contrast, dust particles collected in Beijing, China, during an Asian dust storm were primarily composed of clay minerals (35.5wt%, by weight percentage), quartz (30.3wt%), and calcite (14.0wt%), followed by feldspar (8.7wt%), pyrite (1.0wt%), and hornblende (0.4wt%), along with noncrystalline materials (10.1wt%) (Shi et al., 2005).

Panta et al. (2023) conducted detailed field measurements using electron microscopy in the Sahara Desert of Morocco, reporting the statistical characteristics of the single-particle composition, size, mixing state, and aspect ratio of newly emitted mineral dust. Kok et al. (2023) also highlight that dust-snow interactions generate a global annual-mean radiative forcing of $+0.013$ W m$^{-2}$ (90% confidence interval: 0.007–0.03 W m$^{-2}$), with large uncertainties primarily attributed to variations in dust-snow mixing state, particle size distribution, and chemical composition. To date, no studies have comprehensively analyzed the composition, size, and morphology of dust in snow or clarified the interrelationships among these characteristics. This lack of understanding significantly limits accurate assessments of the optical properties and radiative effects of dust in ice and snow (Flanner et al., 2021; He et al., 2024).

Based on a field snow observation experiment conducted in Changchun, northeastern China, in November 2020, this study utilized intelligent scanning electron microscopy with an energy-dispersive X-ray analyzer to investigate in detail the composition, size, and morphological characteristics of dust during dry and wet deposition. These statistically significant parameters were subsequently used to constrain the complex refractive index and optical absorption inversion of dust, providing more accurate dust optical parameter inputs for snow radiative transfer models, and enhancing the accuracy of climate effect assessments of dust in snow.

**2 Methods**

**2.1 Snow sample collection and analysis**

Our previous study has detailed the snow field experiment conducted in Changchun (Wang et al., 2023). The sampling site is located at the meteorological station of Lvyuan District (43°88′N, 125°25′E), with no apparent sources of air pollution emissions in the visual range. During and after a heavy snowfall from November 19 to December 17, 2020, we collected snow samples every two days, yielding a total of one fresh snowfall sample (wet deposition) and 15 aged surface snow samples (dry and wet deposition). This study selected five samples for measurement and analysis at intervals of 6-8 days, including one wet deposition sample (D1) and four dry/wet deposition samples (D7, D15, D23, and D29; "D" denotes days). Briefly, the selected snow samples were melted at room temperature, and an appropriate volume of the snow solution was taken based on the cleanliness of the snow sample (20 ml for D1 and 1 ml for the rest four samples). The solution was filtered through a polycarbonate membrane with a diameter of 25 mm and a pore size of 0.1 μm to separate the particles. The membrane was then transferred to a storage box and dried in a desiccator. Prior to analysis, a filter membrane approximately 0.5 cm² was cut and gold-plated. The samples were placed in the electron microscope sample chamber for vacuum processing, and data were collected and analyzed using the Environmental Particle Analysis Software (IntelliSEM-EPAS$^{TM}$) of the intelligent scanning electron microscope.

The IntelliSEM-EPAS$^{TM}$ system automatically scans multiple matrix areas within the field of view. By collecting backscattered signals from the scanning electron microscope (TESCAN Mira3) and comparing the image signal intensity with preset threshold levels, particles are detected. Upon detection, the system automatically

records the morphology images and positions of the particles on the polycarbonate membrane and utilizes two Bruker XFlash 6|60 energy dispersive spectroscopy (EDS) detectors to analyze the relative content of 24 chemical elements (C, O, Na, Mg, Al, Si, P, S, Cl, K, Ca, Ti, V, Cr, Mn, Fe, Co, Ni, Cu, Zn, Sn, Ba, Se, and Pb) in the particles. This process rapidly generates high-definition images and energy spectrum data for each particle (thousands of particles per hour). Additionally, IntelliSEM-EPAS[TM] provides detailed measurements of the maximum and minimum diameters, average diameter, particle projection area, roundness, and aspect ratio with the acquired particle SEM images based on a built-in image processing module (Zhao et al., 2022). Compared to manually operated scanning electron microscope experiments, the IntelliSEM-EPAS[TM] system has the advantages of intelligent control and fast analysis speed, allowing for the acquisition of a large amount of environmental particle information in a short time, including detailed data on particle concentration levels, morphology characteristics, and component content across arbitrary size ranges, and were also comparable to the results from bulk analysis (Peters et al., 2016; Wagner and Casuccio, 2014). The elemental concentrations obtained by CCSEM show good consistency with bulk analysis results from atomic absorption (AA), bulk X-ray fluorescence (XRF), proton-induced X-ray emission (PIXE), and anion chromatography (IC) (Casuccio et al., 1983). Mamane et al. (2001) also showed that 360 particles were sufficient to obtain representative results in CCSEM analysis of particle types and size distributions, based on comparisons of 360, 734, 1456, and 2819 individual particles. Although CCSEM has a superior advantage in high efficiency for

measuring large quantities of particles, it encounters challenges with certain types of

particles that have complex morphologies, such as soluble salts and soot (Peters et al.,

2016). CCSEM-induced errors may include particle overlap, contrast artifacts, sizing

inaccuracies, and particle heterogeneity (Mamane et al., 2001). Consequently, manual

error correction is typically performed prior to data processing.

**2.2 Dust microphysical properties derived from IntelliSEM-EPAS$^{TM}$**

Based on the IntelliSEM-EPAS$^{TM}$ system, this study obtained the geometric

information and energy spectrum data of about 4,000-5,000 particles in each sample,

aiming to reveal the statistical characteristics of the microphysical properties of

insoluble particles in snow. Specifically, according to Kandler et al. (2007), particles

with a relative mass proportion of C and O elements exceeding 95% were roughly

classified as carbonaceous particles. Then, for all remaining particles, the elemental

index of each element other than C and O was calculated. Based on single-particle

composition quantification, the elemental index of element X is defined as the atomic

ratio of the concentration of the considered element to the sum of the concentrations of

the quantified elements (Panta et al., 2023).

$$|\mathbf{X}| = \frac{X}{\substack{(Na+Mg+Al+Si+P+S+Cl+K+Ca+Ti+V+Cr \\ +Mn+Fe+Co+Ni+Cu+Zn+Sn+Ba+Pb)}} \tag{1}$$

The elemental symbol indicates the relative contribution measured for each particle (in

atomic percent). Using the obtained elemental indices and combining K-Means

clustering algorithms and manual experience, these non-carbonaceous particles were

classified (Kandler et al., 2007; Panta et al., 2023; Zhao et al., 2022). The main principle

of the K-means clustering algorithm is to use the k-means algorithm to classify particles

with similar chemical compositions into 30 types based on the elemental index of each

element, and then, according to relevant research and manual experience classification

principles of EDS spectra (Panta et al., 2023), classify the 30 types into 12 mineral

phases by merging some similarly classified clusters, with particle categories named

after their most common chemical composition, including quartz-like, hematite-like,

rutile-like, kaolinite-like, chlorite-like, illite-like, hematite-like, clay-minerals-like etc.

Figure S1 presents the percentage distribution of elemental indices (excluding C and O)

for 12 categories of mineral particles. Specifically, hematite-like, quartz-like, rutile-like,

apatite-like, and dolomite-like particles are predominantly characterized by Fe, Si, Ti,

Ca, and Mg, respectively. Kaolinite-like particles are enriched in Al and Si, while clay

mineral-like and Ca-rich silicate particles contain significant amounts of Al and Si,

along with notable Ca content, with the latter exhibiting a higher Ca concentration. In

contrast, illite-like, smectite-like, and chlorite-like particles, in addition to being

enriched in Al and Si, also contain varying amounts of K, Mg, and Fe, respectively.

Correspondingly, representative SEM images of particles are presented within each

mineral category panel.

The size distribution of different types of particles is described using a normal

distribution, specifically expressed as (Flanner et al., 2021; Li et al., 2021):

$$n_r = \frac{dN}{dr} = \sum_{i=1}^{n} \frac{N_i}{\sqrt{2\pi} r \ln(\sigma_i)} \exp\left\{ -\frac{1}{2} \left[ \frac{\ln(r) - \ln(r_i)}{\ln(\sigma_i)} \right]^2 \right\} \qquad (2)$$

where $N_i$ is the total number of particles per unit volume in the i-th size mode, $r_i$ is

the mean radius, and $\sigma_i$ is the geometric standard deviation. These parameters can be

fitted from the measured data. Similarly, the aspect ratio (AR) of particles is also

expressed as a normal distribution function (Panta et al., 2023):

$$n_{AR} = \frac{dN}{dAR} = \sum_{i=1}^{n} \frac{N_i}{\sqrt{2\pi}AR\ln(\sigma_i)} \exp\left\{-\frac{1}{2}\left[\frac{\ln(AR)-\ln(AR_i)}{\ln(\sigma_i)}\right]^2\right\} \tag{3}$$

**2.3 Dust light absorption and snow albedo calculation**

Based on the proportion of different mineral phases in the dust, the effective volume refractive index ($m_{eff}$) of mineral mixtures in snow aerosols was calculated using the effective medium approximation (EMA) method. Specifically, for binary mixtures, the effective complex refractive index under EMA-Bruggeman approximation can be written as (Kahnert, 2015):

$$m_{eff} =$$

$$\sqrt{\frac{1}{4}[m_1^2(2-3f) + m_2^2(3f-1)] + \sqrt{\left[\frac{1}{16}[m_1^2(2-3f) + m_2^2(3f-1)]^2 + \frac{1}{2}m_1^2m_2^2\right]}} \tag{4}$$

where $m_1$ is the complex refractive index of the background matrix, $m_2$ is the complex refractive index of the inclusions, and f is the volume fraction of the inclusions. The effective complex refractive index for multicomponent mixtures can be obtained by repeating the above process. The refractive indices of different minerals used in this study were obtained from the spectral refractive index dataset of the main mineral components and chemical compositions provided by Zhang et al. (2024). For more detailed information about the dataset, refer to Zhang et al. (2024). Subsequently, using the effective complex refractive indices of dust constrained by observations, size distribution, and aspect ratio (AR) data, we calculated the mass absorption coefficient, single scattering albedo, and asymmetry factor of different types of dust particles using

the MOPSMAP program package (Gasteiger and Wiegner, 2018).The MOPSMAP

model is a comprehensive aerosol optical property model combining T-matrix, Mie

scattering theory, and geometric optics, widely used in calculating complex aerosol

optical parameters (Kanngiesser and Kahnert, 2021; Shi et al., 2022b).

The simulation of snow albedo was executed by our team's developed the Spectral

Albedo Model for Dirty Snow (SAMDS) (Wang et al., 2017), which has been applied

in many studies and is applicable to semi-infinite snow depth scenarios (Shi et al., 2021;

Li et al., 2021). Its accuracy is also well validated, achieving an albedo accuracy of

±0.02 compared to field spectroradiometer data (Wang et al., 2017). Specifically, the

albedo of a snow-covered field containing dust under clear sky conditions can be

expressed as:

$$R_d(\lambda) = \exp\left(-4\sqrt{\frac{8\pi B R_{ef} k(\lambda)}{9\lambda(1-g)} + \frac{2\rho_{ice} R_{ef}}{9(1-g)} MAC_{Dust} \cdot C_{Dust}} \cdot \frac{3}{7}(1 + 2\cos(v_0))\right)$$

(5)

where $\lambda$ is the wavelength in μm; $v_0$ is the solar zenith angle; $k(\lambda)$ is the imaginary

part of the complex refractive index of ice. $\rho_{ice}$ and $R_{ef}$ represent the density and

effective radius of snow grains (in μm), respectively; g is the asymmetry factor of snow

grains (weighted average of the scattering angle cosine); B is a factor related only to

the shape of the snow grains. $MAC_{Dust}$ is the mass absorption coefficient of dust, and

$C_{Dust}$ is the concentration of dust particles in the snow. SAMDS uses 480 bands (0.2–

5.0 μm) to resolve spectral albedo. Here we used B = 1.27 and g = 0.89 to characterize

spherical snow grains (Wang et al., 2017), SAMDS is also capable of simulating the

albedo of non-spherical snow grains, and our previous work has explored the albedo

variation induced by snow grain shape (Shi et al., 2022a), which will not be reiterated here. Additionally, this study assumes dust-snow external mixing. However, it is worth noting that some studies have indicated that internal mixing can further enhance the dust-induced albedo reduction caused by 5%–30% (He et al., 2019; Shi et al., 2021). Therefore, this assumption may underestimate the impact of dust on albedo.

## 3 Results

### 3.1 The composition of dust in seasonal snow

The composition of dust determines its complex refractive index, which is crucial for studying the radiative effects of dust (Reynolds et al., 2020; Lee et al., 2020). This study identified a total of 12 mineral components, including hematite-like, quartz-like, rutile-like, clay-mineral-like, illite-like, kaolinite-like, smectite-like, chlorite-like, apatite-like, Ca-rich silicates, domolite-like, and others. However, it is important to handle this classification scheme with caution, as each particle may consist of different minerals, which may have variable or ambiguous compositions. Therefore, the groups used cannot uniquely identify minerals but rather indicate the most likely minerals matching the particle composition. This is reflected in the suffix "-like" used in the group naming scheme. Given the existence of other potential identification methods, each with its own advantages and limitations, the complete dataset generated and used in this study can be utilized for future research. Figure 1 (Figure S2) shows the number (mass) relative proportions of different mineral components in dry and wet deposition snow samples at different size resolutions, indicating significant trends observed among different

particle groups with changes in size categories. For all samples, kaolinite-like is the most abundant, present in all size ranges, with its abundance decreasing with increasing size. Quartz-like particles have nearly similar abundance in each size category (approximately 10%-20%), which is higher than the values reported by Panta et al. (2023) for dust from Morocco (approximately 5%). Similarly, clay-minerals-like are evenly distributed across each size category, accounting for about 4% of the relative abundance. Hematite-like exhibits similar relative abundances, but its contribution decreases with increasing particle size, and its strong light-absorbing properties have drawn widespread attention (Li et al., 2024; Zhang et al., 2015; Moteki et al., 2017). In contrast, chlorite-like's relative contribution increases with increasing size, with an average abundance of approximately 20%. It is noteworthy that the relative abundance of illite-like is higher in wet deposition samples than in dry deposition samples, possibly due to K-rich illite, considered one of the most effective ice nucleation sources found among different mineral components in dust (Atkinson et al., 2013; Harrison et al., 2022). Additionally, the relative abundance of quartz-like in dry deposition samples is significantly lower than in wet deposition samples, which is closely related to the migration process of quartz-like particles in snow. Table S1 provides the relative proportions of different mineral components within the measured size range (0.2-10 µm). Overall, dust in Changchun snow is primarily composed of kaolinite-like (36%), chlorite-like (19%), quartz-like (15%), illite-like (14%), hematite-like (5%), and clay-minerals-like (4%) and other components. In comparison, Shi et al. (2005) reported mineralogical properties of Asian dust primarily consist of clay minerals (35.5wt%, by

weight percentage), quartz (30.3wt%), and calcite (14.0wt%), followed by feldspar

(8.7wt%), pyrite (1.0wt%), and hornblende (0.4wt%). For the Middle East, Prakash et

al. (2016) reported relative mass abundances of clay minerals ranging from 45% to 75%,

plagioclase from 5% to 54%, and quartz from 0.1% to 10.2% as major components.

Considering that industrial activities (e.g., coal combustion, urban construction, and

road dust) emit quartz-rich particles, while long-range transport from arid regions (e.g.,

the Gobi Desert) contributes illite, which is consistent with the dust profile in Asia (Li

et al., 2021). The anthropogenic contribution (e.g., hematite-like particles) aligns with

the presence of nearby steel production facilities. Therefore, our results suggest that

dust is likely a mixture of local and long-range sources.

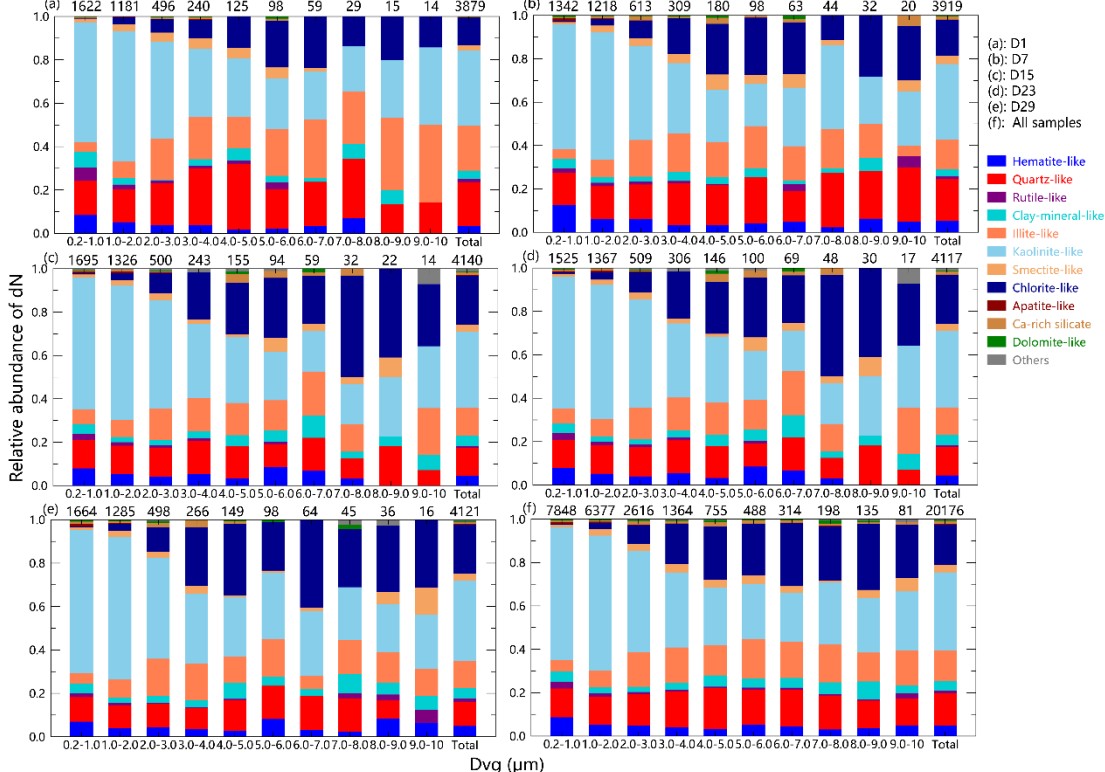

**Figure 1.** Size-resolved number abundance of different particle groups for D1 sample

(a), D7 sample (b), D15 sample (c), D23 sample (d), D29 sample (e), and All samples

(f). The numbers on top represent total particle counts in the given size bin.

**3.2 Size distribution and aspect ratio of dust in seasonal snow**

Particle size is a key factor influencing the light-absorbing properties of dust, which has received widespread attention in field observations, numerical models, and satellite remote sensing (Castellanos et al., 2024; González-Flórez et al., 2023; Song et al., 2022). Figure 2a illustrates the size distribution characteristics of dust particles collected from snow samples at different periods, indicating that the peak particle size of dust during dry deposition did not vary significantly. All samples exhibited similar size distributions, with geometric mean radii ranging from 0.35 to 0.37 µm and geometric standard deviations from 1.88 to 2.12, comparable to findings reported in other studies (Kok, 2011; Di Mauro et al., 2015; Kok et al., 2017). Interestingly, significant differences in size spectra were observed among different mineral components (Figure S3 and Table S2), considering only the cases where the fitted values passed significance tests. Chlorite-like particles exhibited the coarsest size spectrum (median radius = 1.32 µm), nearly double that of smectite-like particles (0.57 µm), likely due to their tendency to aggregate during atmospheric transport (Formenti et al., 2014). Illite-like particles displayed the widest size range (0.38-0.59 µm) across different snow samples, possibly reflecting multiple source regions or differential atmospheric processing. The dominant kaolinite-like and quartz-like particles shared similar size distributions centered around 0.36 µm, consistent with their common origin in soil fragmentation (Kok, 2011), though kaolinite exhibited slightly less size variability. Together these components represented 51% of particles and primarily determined the overall dust size characteristics. Particularly noteworthy were hematite-

like particles, which despite being the smallest at 0.29 µm characteristic of iron oxide

condensation formation, disproportionately influenced radiative properties due to their

exceptional light absorption (Formenti et al., 2014; Go et al., 2022).

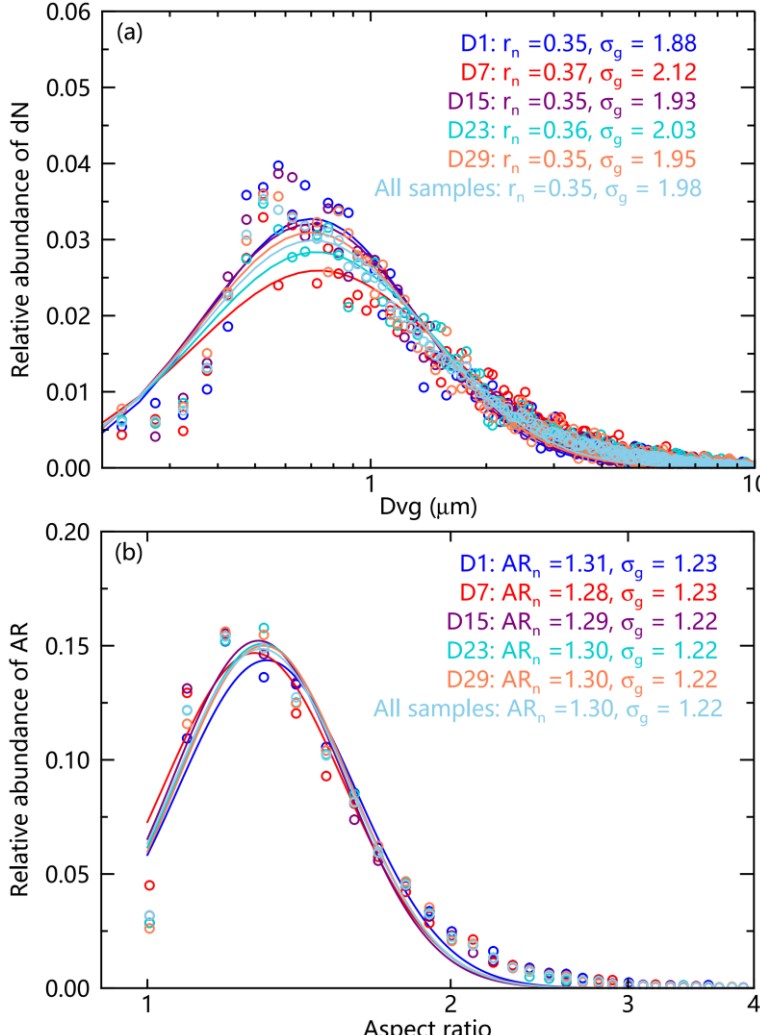

**Figure 2.** Relative abundances of (a) logarithmic dust size number distributions dN/

(dlogD$_p$) and (b) logarithmic dust AR number distributions dN/ (dlogAR) for different

snow samples. Dvg: particle diameter of dust in snow, r$_n$: the number median radius,

$\boldsymbol{\sigma_g}$: the geometric standard deviation.

Aspect ratio (AR) is another critical geometric parameter of dust particle that affects

their light-absorbing properties (Botet and Rai, 2013; Haapanala et al., 2012; Huang et

al., 2023). Figure 2b describes the spectral distribution of aspect ratios of dust particles in dry and wet deposition samples. Similar to the size results, the aspect ratio of dust particles during dry and wet deposition did not show significant variations, with all samples displaying similar spectral distributions. The geometric mean values ranged from 1.28 to 1.31, with geometric standard deviations from 1.22 to 1.23. These results are slightly lower than those reported in atmospheric dust studies, such as measurements of dust from Morocco and Asia with AR values of 1.46 and 1.40, respectively (Kandler et al., 2009; Okada et al., 2001). During the Fennec campaign in central Sahara, a median AR of 1.3 was found (Rocha-Lima et al., 2018), and measurements of dust particles collected in the Sahara air layer and marine boundary layer during the AERosol Properties-Dust (AER-D) period showed median AR values of 1.30–1.44 for particles ranging from 0.5 to 5 µm and 1.30 for particles from 5 to 10 µm, and 1.51 for particles from 10 to 40 µm (Ryder et al., 2018). Furthermore, we also explored the spectral characteristics of aspect ratios of different mineral components (Figure S4 and Table S3). Unlike the size distribution, although there are differences in aspect ratios among different components, the variation range is not large. Most mineral component groups have similar median AR values of 1.30, except for hematite and clay minerals, which have the lowest median AR of 1.27 and the highest median AR of 1.37, respectively. The AR of the same mineral component group shows no significant differences among different samples. Additionally, we found that AR is generally independent of particle size and type (Figure S5), consistent with the results of Panta et al. (2023).

**3.3 Dust light absorption and its effects on snow albedo**

The refractive index of various mineral components exhibits significant variation. Figure S6 illustrates the complex refractive indices (both real and imaginary parts) of the eight principal mineral component groups identified in this study. The imaginary parts, indicative of absorption, vary by up to six orders of magnitude. Hematite shows the highest imaginary part of the complex refractive index, indicating the strongest light-absorbing properties, while quartz displays the smallest, indicating the weakest. The complex refractive indices of kaolinite, illite, chlorite, and smectite present relatively similar values, suggesting minimal variation in their light-absorbing properties. Based on the complex refractive index database of mineral component groups and combined with volume relative proportions under observational constraints, an effective medium approximation method is used to obtain the effective complex refractive index of dust in snow. Additionally, to assess the impact of different mineral component groups on the effective complex refractive index, we adjusted the initial volume proportions of hematite, kaolinite, chlorite, and illite by factors of 1.25, 1.50, 1.75, and 2.0, respectively, while keeping the relative proportions of other components unchanged, and finally normalizing the proportions of all components. Figure 3 illustrates the variation in the effective complex refractive index of dust with wavelength under these scenarios, focusing on the imaginary parts related to absorption. Overall, $k_{dust}$ is distributed within a narrow range (~0.001–0.01), gradually decreasing with increasing wavelength in the UV and VIS bands, and then stabilizing in the NIR band, comparable to values reported in other literature. Notably, an increase in the

relative proportion of hematite leads to a significant rise in $k_{dust}$, especially in the visible

spectrum. Conversely, increases in the relative proportions of kaolinite, chlorite, and

illite cause a slight decrease in $k_{dust}$, due to the reduced relative proportion of hematite

after normalization.

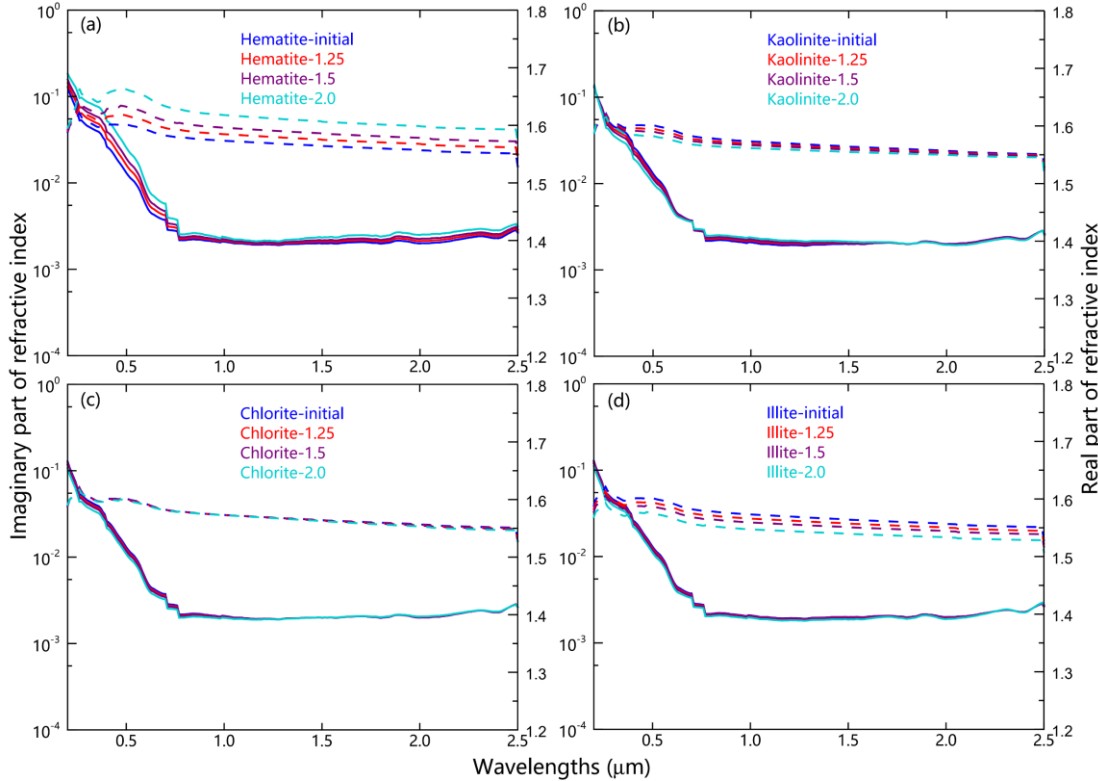

**Figure 3.** Complex spectral refractive indices of dust mixtures in scenarios with

different composition group percentages. The solid and dashed lines in the diagram

represent the imaginary and real parts, respectively. The default average volume

fraction of each mineral group is 35.6% Kaolinite, 19.4% Chlorite, 15.2% Quartz, 14.6%

Illite, 4.5% Hematite, 3.1% Smectite, and 1.1% Rutile. (a), (b), (c), and (d) represent

the effects of changes in the proportion of hematite, kaolinite, chlorite, and illite,

respectively.

Furthermore, incorporating observed dust size distribution and AR spectra

characters, we calculated the mass absorption cross-section ($MAC_{dust}$), as shown in Figure 4. Similar to $k_{dust}$, $MAC_{dust}$ is distributed within a narrow range (~0–0.3 m²/g), gradually decreasing with increasing wavelength in the UV and VIS bands, and approaching stability (~0) at wavelengths greater than 1000 nm. An increased relative proportion of hematite enhances $MAC_{dust}$ in the visible spectrum. For instance, doubling the relative proportion of hematite raises $MAC_{dust}$ at 500 nm from 0.14 m²/g to 0.19 m²/g. However, changes in the relative proportions of kaolinite and chlorite have minimal effects on $MAC_{dust}$, consistent with the results for $k_{dust}$. Additionally, an increase in $R_{dust}$ significantly reduces $MAC_{dust}$ in the UV and VIS bands, weakening its spectral dependence. For example, when $R_{dust}$ is increased by factors of 1.25, 1.5, and 2.0, $MAC_{dust}$ at 300 nm decreases by 20% (0.20 m²/g), 33% (0.17 m²/g), and 48% (0.13 m²/g), respectively, and at 500 nm, it decreases by 12% (0.12 m²/g), 21% (0.11 m²/g), and 34% (0.09 m²/g). Overall, the measured $MAC_{dust}$ values (0–0.3 m²/g) show regional variations that reflect compositional differences: while comparable to Saharan dust (0.1–0.25 m²/g, Balkanski et al., 2007), they are significantly lower than Tibetan Plateau dust (0.3–0.5 m²/g, Li et al., 2021) and slightly higher than Colorado (San Juan Mountains) dust (0.05–0.15 m²/g, Skiles et al., 2017). This pattern correlates with hematite content, decreasing from 8–12% in Tibetan Plateau dust to 5% in our samples and just 2–3% in Greenland dust (Polashenski et al., 2015). The distinct quartz-rich signature in our samples (15% vs <5% in other regions) may reflect unique industrial emission sources in northeastern China.

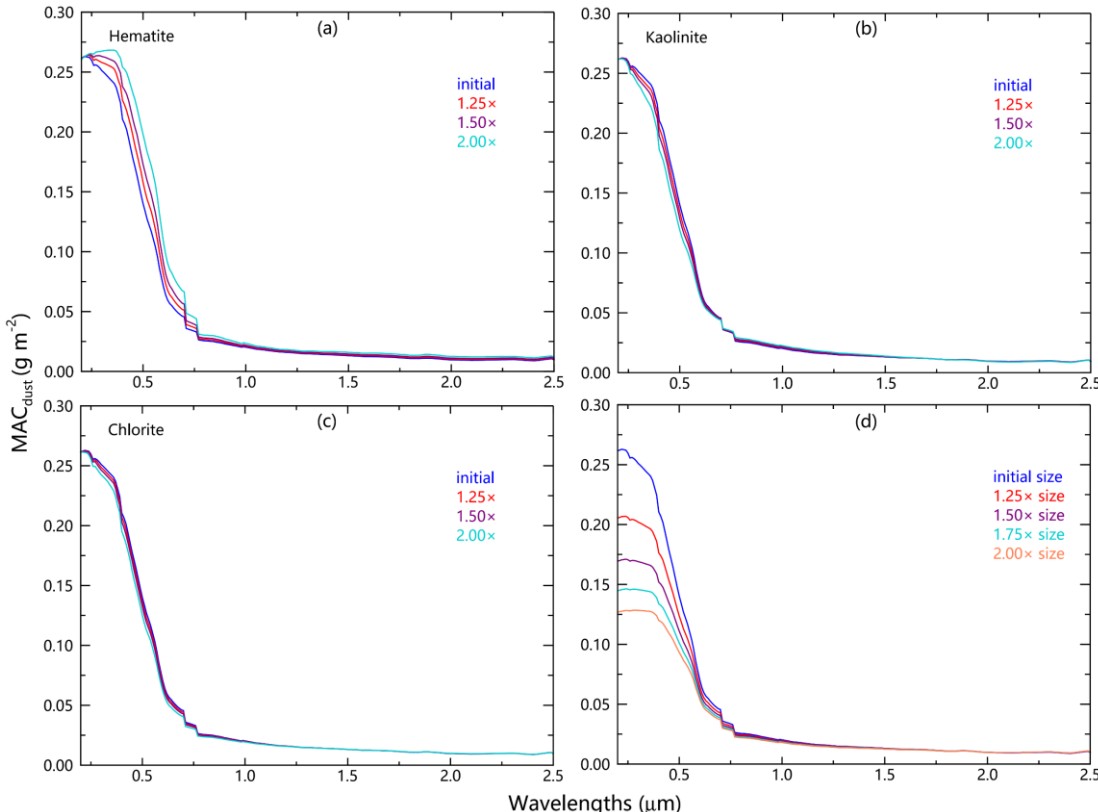

**Figure 4.** Spectral variations in the dust mass absorption cross-sections (MACs) for

different simulation scenario: (a) Hematite, (b) Kaolinite, (c) Chlorite, and (d) Size.

Here the dust aspect ratio is fixed at 1.3.

Figure 5a illustrates the impact of changes in the relative proportion of hematite on the

spectral snow albedo, considering scenarios with low, medium, and high dust loads in

snow, assuming a snow particle size of 500 μm (medium scenario). It can be observed

that changes in spectral albedo due to variations in dust concentration and composition

proportions generally occur in the visible light spectrum, while the near-infrared (NIR)

spectrum is primarily influenced by the microphysical properties of snow particles

themselves (Gardner and Sharp, 2010; He and Flanner, 2020), thus unaffected by dust

concentration and composition proportions. Specifically, spectral albedo decreases in

the UV and visible light (UV-Vis) bands with increasing dust concentration, with a

further decrease observed with rising proportions of hematite. Similar to Figure 5a, Figure 5b describes changes in spectral albedo of snow under different dust particle sizes, showing that increasing dust particle size can mitigate the decline in spectral albedo in the visible light spectrum, which is more pronounced in high dust load scenarios. For example, doubling the dust particle size increases the spectral albedo (300 nm) from 0.946, 0.840, and 576 to 0.961, 0.882, and 0.673 for dust concentrations of 1, 10, and 100 ppm in snow, respectively. Figures 5c and 5d respectively illustrate the effects of changes in the relative proportion of hematite and dust particle size on the reduction in snow albedo, considering three snow particle size scenarios. Specifically, the reduction in albedo increases with increasing dust concentration and snow particle size, further exacerbated by an increase in the proportion of hematite, especially in high dust concentration and snow particle size scenarios. Conversely, an increase in dust particle size reduces the reduction in albedo, and increases in dust concentration and snow particle size can further amplify this effect. For instance, in low (high) snow particle size scenarios, increasing the proportion of hematite increases the reduction in albedo caused by dust concentrations of 1, 10, and 100 ppm in snow from 0.007 (0.022), 0.028 (0.084), and 0.099 (0.257) to 0.008 (0.026), 0.033 (0.098), and 0.115 (0.291). Conversely, increasing the dust particle size reduces the reduction in albedo caused by dust concentrations of 1, 10, and 100 ppm in snow to 0.005 (0.017), 0.022 (0.066), and 0.081 (0.217). These results emphasize the complex effects of dust composition, particle size, concentration, and snow particle size on snow albedo.

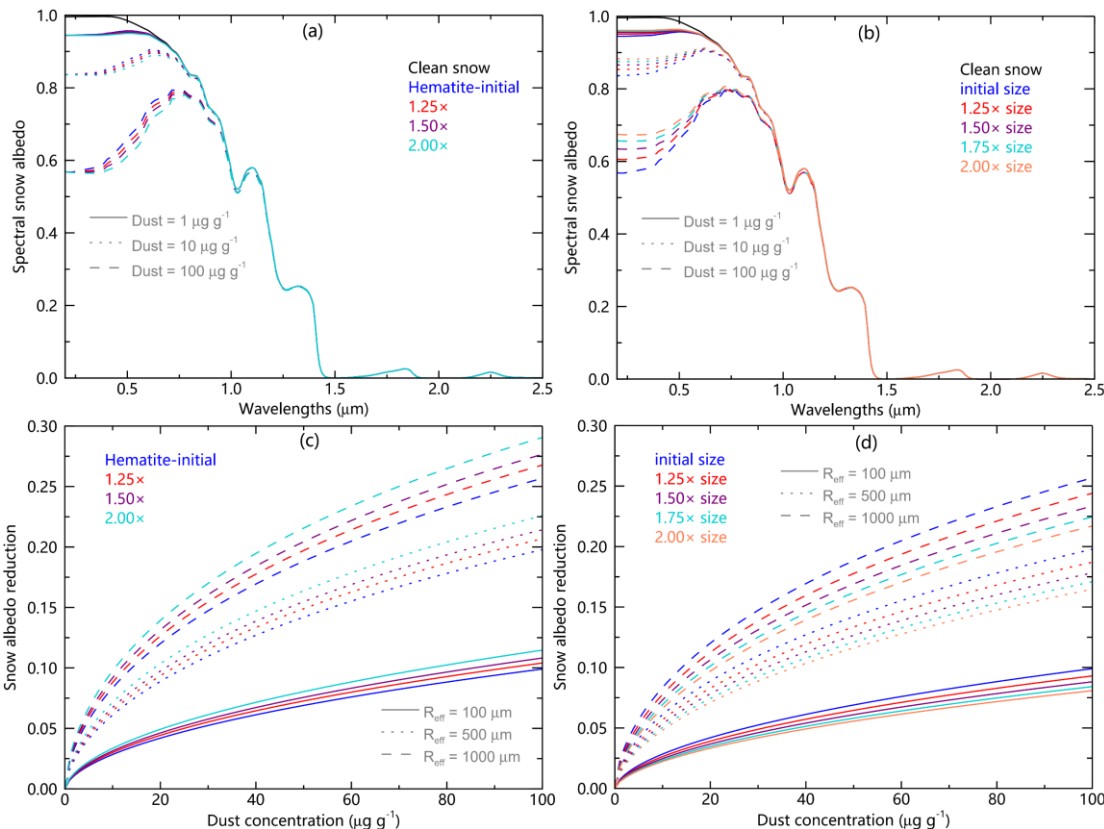

**Figure 5.** (a) Spectral snow albedo in the wavelength range of 0.2–2.5 μm for different

dust concentrations and hematite percentages, with assumed snow radii of 500μm. (b)

Spectral snow albedo for different dust concentrations and sizes. (c) Broadband snow

albedo reduction as a function of dust concentration for different hematite percentages

and snow snow-grain radii (100, 500, and 1,000 μm). (d) Similar to (c), but hematite

percentage is replaced with dust size.

**4 Summary and discussion**

This study employed CCSEM technology to quantitatively analyze insoluble

particulate matter in snow in Changchun, ranging from 0.2 to 10 μm, and identified 12

mineral component groups through K-means cluster analysis and empirical

identification. The findings indicate that the dust in Changchun snow primarily

comprises kaolinite-like (36%), chlorite-like (19%), quartz-like (15%), illite-like (14%),

hematite-like (5%), and clay-minerals-like (4%), with no significant changes in the

proportions of different mineral components during dry deposition processes. In

contrast, wet deposition samples contain higher proportions of illite and quartz, which

may be attributed to illite as an effective source of ice nuclei and the dynamic migration

of quartz in snow. The study also found that the size and aspect ratio (AR) of dust follow

normal distribution characteristics, with geometric means and standard deviations of

0.35–0.37 μm, 1.88–2.12 for size, and1.28–1.31, 1.22–1.23 for AR, respectively.

Although there were no significant changes in the size and AR of dust during dry and

wet deposition processes, significant variability was observed among different mineral

component groups in terms of size and AR. Subsequently, based on statistically derived

characteristics of dust components, size, and AR under observational constraints, we

analyzed the light absorption characteristics of dust. The mass absorption cross-section

($MAC_{dust}$) was found to be distributed within a narrow range ($\sim 0$–$0.3$ m²/g). An increase

in the relative proportion of hematite was observed to increase $MAC_{dust}$, while an

increase in dust particle size decreased $MAC_{dust}$ by a specific percentage (10%–50%).

Finally, the study discussed the complex effects of dust composition, particle size,

concentration, and snow particle size on snow albedo. The results indicate that an

increase in the relative proportion of hematite further enhances the reduction in snow

albedo caused by dust, whereas an increase in dust particle size mitigates this reduction.

Additionally, increases in dust concentration and snow particle size can further amplify

these effects.

Compared with bulk sample collection and other techniques, we emphasize that

CCSEM technology provides an innovative approach to detect the statistical characteristics of mineral composition, size distribution, and shape (AR) of dust in snow, significantly enhancing the accuracy of dust radiative forcing in model simulations. However, it is worth noting that although mineralogy provides strict definitions for mineral phases based on composition and crystal structure, atmospheric dust particles typically consist of heterogeneous mixtures. Currently, the scientific community lacks standardized protocols for classifying the mineralogical components of such complex particulate assemblages, making it difficult to compare dust composition reported in different literature, severely limiting research on dust chemical composition in different regions globally (Castellanos et al., 2024; Zhang et al., 2024). Therefore, we call for the establishment of strict criteria for distinguishing mineral components as soon as possible, which will also support high-spectral projects and space programs developed and implemented by international societies and aerospace institutions to enhance understanding of mineral composition in terrestrial dust source regions (Green et al., 2020; Guanter et al., 2015). On the other hand, there is still a lack of understanding of the basic mineralogical and physical properties of dust particles, including key minerals such as hematite and goethite's spectral refractive indices. Measurements of hematite refractive indices currently vary widely, hindering attempts to calculate dust optical properties and forcing changes (Zhang et al., 2024). In addition, the irregular shapes of dust particles cannot be represented by simple mathematical models, and the lack of comprehensive and realistic shape models is a prominent issue in dust optical modeling, distinguishing it from other aerosol types (Huang et al., 2023; Ito et al., 2021). Overall,

the greatest limitation lies in the lack of detailed, region-specific, statistically representative information on the microphysical properties of base dust particles — size distribution, morphology, complex refractive index spectra, heterogeneity of internal structures, and resulting optical characteristics.

**Supporting Information**

Figures S1−S6.

Tables S1-S3

**Data availability statement**

The data used for analysis are available via a Zenodo archive, which can be found in the references (https://zenodo.org/doi/10.5281/zenodo.14633496, last access: 12 Jan 2025).

**Author contributions**

X.W. and J.W. designed the study and evolved the overarching research goals and aims. T.S. wrote the first draft with contributions from all co-authors. T.S., Z.W., Y.Z. and W.P. collected snow samples and performed sampling analyses. T.S. and J.C. applied formal techniques such as statistical, mathematical and computational to analyze study data. Y.B. and Z.H. provided the majority of the methodology and software. The other authors provided technical guidance. All authors contributed to the improvement of results and revised the final paper.

**Competing interests**

The authors declare that they have no conflict of interest.

**Financial support**

This research is jointly supported by the National Science Fund for Distinguished

Young Scholars (42025102), the Postdoctoral Fellowship Program of China

Postdoctoral Science Foundation (GZC20230674), the National Natural Science

Foundation of China (42375068, 42301142 and 42405099), and the Natural Science

Founds of Gansu Province, China (21ZDKA0017).

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
