# Peer review of "Insights into microphysical and optical properties of"

_EGUsphere, 2025_

## Author Comment (AC1)

Dear Reviewers,

Thank you for taking the time to review this manuscript. We really appreciate the reviewers' comments, which have helped us to improve the paper quality substantially. We have addressed all the comments very carefully in our following point-by-point responses. Our responses start with "R:".

General comments

In this study, mineral dust particles from snowpack were analyzed by SEM. They classified the measured particles based on their compositions and discussed the effects on their optical properties. This study is generally well designed and the topic is interesting. However, I have a concern with the classification of the particles and the subsequent discussions based on the classification. I suggest having more description of the classification and providing SEM chemical and image data of the particles that they classified.

R: We sincerely appreciate the reviewer's constructive feedback. To address concerns about particle classification and enhance transparency, we added Figure S1, which shows the percentage of each elemental index (without C and O) and the corresponding SEM images of typical particles for the 12 categories of mineral particles (See Page 10, Line7-16).

*"Figure S1 presents the percentage distribution of elemental indices (excluding C and O) for 12 categories of mineral particles. Specifically, hematite-like, quartz-like, rutile-like, apatite-like, and dolomite-like particles are predominantly characterized by Fe, Si, Ti, Ca, and Mg, respectively. Kaolinite-like particles are enriched in Al and Si, while clay mineral-like and Ca-rich silicate particles contain significant amounts of Al and Si, along with notable Ca content, with the latter exhibiting a higher Ca concentration. In contrast, illite-like, smectite-like, and chlorite-like particles, in addition to being enriched in Al and Si, also contain varying amounts of K, Mg, and Fe, respectively. Correspondingly, representative SEM images of particles are presented within each mineral category panel."*

[Figure]

**Figure S1.** The percentage of each elemental index (without C and O) for the 12 categories of mineral particles. Subplots (a)-(l) represent results for hematite-like, quartz-like, rutile-like, clay-mineral-like, illite-like, kaolinite-like, smectite-like, chlorite-like, apatite-like, Ca-rich silicates, domolite-like, and others, respectively. Correspondingly, representative SEM images of particles are presented within each mineral category panel. The red circle and whiskers denote the average value and mean deviation. The data for each particle is shown as gray solid dots.

Specific comments

Page 1, line 2: "industrial-polluted snowpack" Throughout the manuscript, there is not much discussion and results about the influence of industrial pollution. In an industrially polluted city, there should be many anthropogenic pollutants, but no results are provided. Please provide any results and discussion on anthropogenic pollutants.

R: The title has been revised to "Insights into microphysical and optical properties of typical mineral dust within urban snowpack via wet/dry deposition in Changchun, Northeastern China" to better reflect the focus on mineral dust rather than industrial pollutants. While the sampling site is in an industrial city, the study primarily analyzes dust components. Future work will explicitly address anthropogenic pollutants (e.g., heavy metals, black carbon). Additionally, we added more discussion about potential sources of dust (e.g., natural and anthropogenic sources), see Page 15, Line 5-10.

*"Considering that industrial activities (e.g., coal combustion, urban construction, and road dust) emit quartz-rich particles, while long-range transport from arid regions (e.g., the Gobi Desert) contributes illite, which is consistent with the dust profile in Asia (Li et al., 2021). The anthropogenic contribution (e.g., hematite-like particles) aligns with the presence of nearby steel production facilities. Therefore, our results suggest that dust is likely a mixture of local and long-range sources."*

Page 2 lines 2-7: I do not think it is appropriate to include the name of commercial products in the first sentence of the abstract, although it is up to the author. The first sentence is long and needs to be checked for grammar.

R: The abstract has been rephrased:

*"This study presents the first compositional analysis of dust in snowpack from a typical Chinese industrial city, utilizing computer-controlled scanning electron microscope combined with K-means cluster analysis and manual experience. The dust is predominantly composed of kaolinite-like (36%), chlorite-like (19%), quartz-like (15%), illite-like (14%), hematite-like (5%), and clay-minerals-like (4%), with minor contributions from other components."*

The details of the IntelliSEM-EPAS™ software are provided in the Methods section.

Page 3, lines 7-10: This sentence is misleading and needs to be revised. Many aerosols are deposited on land and in the ocean.

R: The sentence has been revised as follows:

*"Current research indicates that light-absorbing aerosols in the atmosphere (e.g. black carbon, brown carbon, and dust) are eventually deposited on various surfaces, including snow or glaciers through atmospheric diffusion, transport, and dry/wet deposition processes."*

Page 4, lines 16-18: This sentence is awkward. Perhaps "while" is not needed.

R: Revised as follows:

*"Hao et al. (2023) projected a decrease in black carbon deposition on ice and snow under future emission scenarios, and anticipated that heightened dust emissions and deposition fluxes driven by climate change-induced land use changes…"*

Page 6, line 21: Please describe the sampling location in more detail. Is it near a road or industrial facility?

R: Added as follows:

*"The sampling site is located at the meteorological station of Lvyuan District (43°88'N, 125°25'E), with no apparent sources of air pollution emissions in the visual range."*

Page 7, line 1: Aged surface snow should include both wet and dry deposition.

R: Revised as suggestion.

Page 7, lines 4-6: How much snow did you use for the measurements?

R: Fresh snowfall sample was filtered for about 20 ml, the remaining four samples were filtered for 1 ml. The measurements of the snow samples were added in Line 11, Page 7.

Page 7, line 12: Please provide information about the SEM (e.g., company).

R: Addition:

*"...the scanning electron microscope (TESCAN Mira3)…"*

Page 7, line 18: The selected elements are 29, not 24. Some elements (e.g. Rh) may be misclassified because they are very rare. Please check them from the original SEM data.

R: We sincerely apologize for this error. After carefully reviewing the original SEM data, we have confirmed the analysis of 24 elements (including C, O, Na, Mg, Al, Si, P, S, Cl, K, Ca, Ti, V, Cr, Mn, Fe, Co, Ni, Cu, Zn, Sn, Ba, Se, and Pb), which also revised in the manuscript accordingly.

Page 7 line 21: "thousands of particles per hour" How many seconds did you use for chemical analysis? In general, 1 or 2 seconds for EDS is not enough to qualify the elements.

R: The IntelliSEM-EPAS$^{TM}$ utilizes two Bruker XFlash 6|60 energy dispersive spectroscopy (EDS) detectors to analyze the relative content of 24 chemical elements, enabling high analytical efficiency.

Page 8, line 5: "particle concentration level" Did you measure particle concentrations?

R: The particle concentration can be calculated based on the filtered meltwater volume, membrane area, and scanned area. This study does not report such results, our previous research has reported particle concentration levels (Wang et al., 2023).

Page 9, line 1: "K-means clustering algorithms and manual experience" Although I understand that it is difficult to classify particles based only on the K-means clustering technique, please describe the "manual experience". Also, I strongly suggest showing, for example, average compositions of the particle group (e.g., quartz-like, ...) and representative SEM images of the particles. In the current discussion, there is no such data, and I cannot judge whether the classification worked well or not. Thus, I question whether the particle size distributions are adequate or not, i.e., whether the distributions are similar for all particle types.

R: This study uses the k-means algorithm to classify particles with similar chemical compositions into 30 types. The classification is based on the elemental index of each element. Then, according to relevant research and manual classification principles of EDS spectra (Panta et al., 2023), the 30 types are grouped into 12 mineral phases. This is done by merging some similarly classified clusters. The particle categories are named after their most common chemical composition. These include hematite-like, quartz-like, rutile-like, clay-mineral-like, illite-like, kaolinite-like, smectite-like, chlorite-like, apatite-like, Ca-rich silicates, domolite-like, and others. Specifically, hematite-like, quartz-like, rutile-like, apatite-like, and dolomite-like particles are predominantly characterized by Fe, Si, Ti, Ca, and Mg, respectively. Kaolinite-like particles are enriched in Al and Si, while clay mineral-like and Ca-rich silicate particles contain significant amounts of Al and Si, along with notable Ca content, with the latter exhibiting a higher Ca concentration. In contrast, illite-like, smectite-like, and chlorite-like particles, in addition to being enriched in Al and Si, also contain varying amounts of K, Mg, and Fe, respectively.

To address concerns about particle classification and enhance transparency, we added Figure S1, which shows the percentage of each elemental index (without C and O) and the corresponding SEM images of typical particles for the 12 categories of mineral particles (See Page 10, Line 7-16).

*"Figure S1 presents the percentage distribution of elemental indices (excluding C and O) for 12 categories of mineral particles. Specifically, hematite-like, quartz-like, rutile-like, apatite-like, and dolomite-like particles are predominantly characterized by Fe, Si,*

*Ti, Ca, and Mg, respectively. Kaolinite-like particles are enriched in Al and Si, while clay mineral-like and Ca-rich silicate particles contain significant amounts of Al and Si, along with notable Ca content, with the latter exhibiting a higher Ca concentration. In contrast, illite-like, smectite-like, and chlorite-like particles, in addition to being enriched in Al and Si, also contain varying amounts of K, Mg, and Fe, respectively. Correspondingly, representative SEM images of particles are presented within each mineral category panel."*

Page 11, lines 13-16: Similar to the comment above, please describe how you classify the mineral dust particles into these categories.

R: We have supplemented additional descriptions regarding mineral dust classification in Section 2.2, as detailed in our response to previous comment.

Page 23 lines 12-22: "It is worth noting that there is currently no strict set of criteria in the scientific community for classifying dust mineral components" These discussions are misleading. In mineralogy, there are strict definitions of each mineral phase based on composition and crystal structure. The problem with the mineral dust particles used in this study is that they are mixtures and not simple components, resulting in the difficulty of particle classification.

R: As suggested by the reviewers, we have revised the relevant statements to avoid being misleading, See Page 26, Line 4-8.

*"it is worth noting that although mineralogy provides strict definitions for mineral phases based on composition and crystal structure, atmospheric dust particles typically consist of heterogeneous mixtures. Currently, the scientific community lacks standardized protocols for classifying the mineralogical components of such complex particulate assemblages…"*

References:

Hao, D., Bisht, G., Wang, H., Xu, D., Huang, H., Qian, Y., and Leung, L. R.: A cleaner snow future mitigates Northern Hemisphere snowpack loss from warming, Nat Commun, 14, 6074, https://doi.org/10.1038/s41467-023-41732-6, 2023.

Li, Y., Kang, S., Zhang, X., Chen, J., Schmale, J., Li, X., Zhang, Y., Niu, H., Li, Z., Qin, X., He, X., Yang, W., Zhang, G., Wang, S., Shao, L., and Tian, L.: Black carbon and dust in the Third Pole glaciers: Revaluated concentrations, mass absorption cross-sections and contributions to glacier ablation, Sci Total Environ, 789, 147746, https://doi.org/10.1016/j.scitotenv.2021.147746, 2021.

Panta, A., Kandler, K., Alastuey, A., González-Flórez, C., González-Romero, A., Klose, M., Querol, X., Reche, C., Yus-Díez, J., and Pérez García-Pando, C.: Insights into the single-particle composition, size, mixing state, and aspect ratio of freshly emitted mineral dust from field measurements in the Moroccan Sahara using electron microscopy, Atmos Chem Phys, 23, 3861-3885, https://doi.org/10.5194/acp-23-3861-2023, 2023.

Wang, X., Zhang, C., Shi, T., Zhang, D., Zhao, P., and Zhao, P.: Case Investigation on the Influence of In-Snow Particles' Size and Composition on the Snow Light Absorption and Albedo, Geophys Res Lett, 50, e2023GL103362, https://doi.org/10.1029/2023GL103362, 2023.

---

## Author Comment (AC2)

Dear Reviewers,

Thank you for taking the time to review this manuscript. We really appreciate the reviewers' comments, which have helped us to improve the paper quality substantially. We have addressed all the comments very carefully in our following point-by-point responses. Our responses start with "R:".

The authors measured and analyzed the dust composition and microphysical features based on samples in the snowpack from a typical industrial city in China. They found that size distribution and aspect ratio of the dust did not undergo significant changes during dry and wet deposition but exhibited great variability among the different mineral composition groups. The impact of dust composition on snow albedo effect has been less studied in the past. This study using the observations to constrain dust size distribution and composition provides a useful framework to assess dust-snow albedo effect. Overall, the manuscript is well organized, but there are still some places that need more descriptions and clarifications.

Comments:

Introduction: One important missing reference here is the recent review paper (https://doi.org/10.1038/s43017-022-00379-5) on dust climatic effects, which quantifies the dust-snow radiative effects and uncertainties. This could be discussed as a broad context here for the dust-snow albedo effect problem.

R: The related reference has been added in the introduction section in Line 4-7, Page 6. *"Kok et al., (2023) also highlight that dust-snow interactions generate a global annual-mean radiative forcing of +0.013 W m⁻² (90% confidence interval: 0.007–0.03 W m⁻²), with large uncertainties primarily attributed to variations in dust-snow mixing state, particle size distribution, and chemical composition."*

Section 2.2: (1) I would suggest presenting an example for the SEM images of the dust samples and the energy spectrum demonstrating the signals for each key dust composition elements.

R: We have added Figure S1, which shows the percentage of each elemental index

(without C and O) and the corresponding SEM images of typical particles for the 12 categories of mineral particles (See Page 10, Line 7-16).

*"Figure S1 presents the percentage distribution of elemental indices (excluding C and O) for 12 categories of mineral particles. Specifically, hematite-like, quartz-like, rutile-like, apatite-like, and dolomite-like particles are predominantly characterized by Fe, Si, Ti, Ca, and Mg, respectively. Kaolinite-like particles are enriched in Al and Si, while clay mineral-like and Ca-rich silicate particles contain significant amounts of Al and Si, along with notable Ca content, with the latter exhibiting a higher Ca concentration. In contrast, illite-like, smectite-like, and chlorite-like particles, in addition to being enriched in Al and Si, also contain varying amounts of K, Mg, and Fe, respectively. Correspondingly, representative SEM images of particles are presented within each mineral category panel."*

[Figure]

**Figure S1.** The percentage of each elemental index (without C and O) for the 12 categories of mineral particles. Subplots (a)-(l) represent results for hematite-like, quartz-like, rutile-like, clay-mineral-like, illite-like, kaolinite-like, smectite-like, chlorite-like, apatite-like, Ca-rich silicates, domolite-like, and others, respectively. Correspondingly, representative SEM images of particles are presented within each mineral category panel. The red circle and whiskers denote the average value and mean deviation. The data for each particle is shown as gray solid dots.

(2) Also, it will be useful if the authors could also discuss the uncertainties associated with the SEM-EPAS measurement-analysis system.

R: We added more descriptions about the precision evaluation and potential errors in the SEM-EPAS measurement-analysis system (See Page 8, Line 10-22 and Page 9, Line 1-5).

*"Compared to manually operated scanning electron microscope experiments, the IntelliSEM-EPAS$^{TM}$ system has the advantages of intelligent control and fast analysis speed, allowing for the acquisition of a large amount of environmental particle information in a short time, including detailed data on particle concentration levels, morphology characteristics, and component content across arbitrary size ranges, and were also comparable to the results from bulk analysis (Wagner and Casuccio, 2014;Peters et al., 2016). The elemental concentrations obtained by CCSEM show good consistency with bulk analysis results from atomic absorption (AA), bulk X-ray fluorescence (XRF), proton-induced X-ray emission (PIXE), and anion chromatography (IC) (Casuccio et al., 1983). Mamane et al. (2001) also showed that 360 particles were sufficient to obtain representative results in CCSEM analysis of particle types and size distributions, based on comparisons of 360, 734, 1456, and 2819 individual particles. Although CCSEM has a superior advantage in high efficiency for measuring large quantities of particles, it encounters challenges with certain types of particles that have complex morphologies, such as soluble salts and soot (Peters et al., 2016). CCSEM-induced errors may include particle overlap, contrast artifacts, sizing inaccuracies, and particle heterogeneity (Mamane et al., 2001). Consequently, manual*

*error correction is typically performed prior to data processing."*

(3) It is not very clear how the size distribution and aspect ratio were measured. Is it also derived from SEM images? More descriptions are needed.

R: IntelliSEM-EPAS[TM] provides detailed measurements of the maximum and minimum diameters, average diameter, particle projection area, roundness, and aspect ratio with the acquired particle SEM images based on a built-in image processing module. Further clarification could be found in Page 8, Line 6-9.

Section 2.3: More descriptions of the SAMDS model are needed. For example, is it assuming very deep snowpack (e.g., semi-infinite)? What is the accuracy of this model? Does the model assume dust-snow external mixing as previous studies (e.g., https://doi.org/10.1029/2019MS001737) highlighted the importance of dust-snow internal mixing? How many spectral bands are used in SAMDS? Could the SAMDS handle non-spherical snow grains (I believe so)? If so, maybe a sensitivity test by using a nonspherical snow grain assumption will be very useful to quantify the uncertainty caused by snow grain shape.

R: We have added more description of SAMDS (see Page 12, Lines 5-9 and Page 13, Lines 1-5).

*"The simulation of snow albedo was executed by our team's developed the Spectral Albedo Model for Dirty Snow (SAMDS) (Wang et al., 2017), which has been applied in many studies and is applicable to semi-infinite snow depth scenarios (Shi et al., 2021; Li et al., 2021). Its accuracy is also well validated, achieving an albedo accuracy of ±0.02 compared to field spectroradiometer data (Wang et al., 2017)."*

*"SAMDS uses 480 bands (0.2–5.0 μm) to resolve spectral albedo. Here we used B = 1.27 and g = 0.89 to characterize spherical snow grains (Wang et al., 2017), SAMDS is also capable of simulating the albedo of non-spherical snow grains, and our previous work has explored the albedo variation induced by snow grain shape (Shi et al., 2022a), which will not be reiterated here. Additionally, this study assumes dust-snow external mixing. However, it is worth noting that some studies have indicated that internal*

*mixing can further enhance the dust-induced albedo reduction caused by 5%–30% (He et al., 2019; Shi et al., 2021). Therefore, this assumption may underestimate the impact of dust on albedo."*

Section 3.1: Based on the dust composition in this study and literature, would the authors be able to add a small discussion on potential sources for these dust particles (e.g., local or long-range transported? Anthropogenic or natural dust?)?

R: We have added more discussion on potential sources for these dust particles (see Page 15, Lines 5-10).

*"Considering that industrial activities (e.g., coal combustion, urban construction, and road dust) emit quartz-rich particles, while long-range transport from arid regions (e.g., the Gobi Desert) contributes illite, which is consistent with the dust profile in Asia (Li et al., 2021). The anthropogenic contribution (e.g., hematite-like particles) aligns with the presence of nearby steel production facilities. Therefore, our results suggest that dust is likely a mixture of local and long-range sources."*

Section 3.2: It is interesting to see that different mineral components show large differences in size spectra. Any physical explanation for this?

R: We have added more related physical explanations (see Page 16, Lines 13-22 and Page 17, Lines 1-3).

*"Chlorite-like particles exhibited the coarsest size spectrum (median radius = 1.32 μm), nearly double that of smectite-like particles (0.57 μm), likely due to their tendency to aggregate during atmospheric transport (Formenti et al., 2014). Illite-like particles displayed the widest size range (0.38-0.59 μm) across different snow samples, possibly reflecting multiple source regions or differential atmospheric processing. The dominant kaolinite-like and quartz-like particles shared similar size distributions centered around 0.36 μm, consistent with their common origin in soil fragmentation (Kok, 2011), though kaolinite exhibited slightly less size variability. Together these components represented 51% of particles and primarily determined the overall dust size characteristics. Particularly noteworthy were hematite-like particles, which despite*

*being the smallest at 0.29 μm characteristic of iron oxide condensation formation, disproportionately influenced radiative properties due to their exceptional light absorption (Formenti et al., 2014; Go et al., 2022)."*

Figure 4: How do these MAC_dust values compare with previous literature reported values? It will be useful to know this information. This may reflect some uniqueness of dust in this region.

R: We added detailed descriptions to compare the MAC values reported in previous literature (see Page 21, Lines 13-21).

[revised manuscript text omitted]

---

## Author Response (AR2)

Dear Masashi Niwano,

Thank you for your positive feedback and constructive guidance. We sincerely appreciate the thorough review process and are pleased to hear that our revisions have addressed the referees' concerns.

Regarding your editorial comment about the Greenland dust comparison (P.21, L.19), we will add the following reference to support this statement:

Polashenski, C. M., Dibb, J. E., Flanner, M. G., Chen, J. Y., Courville, Z. R., Lai, A. M., Schauer, J. J., Shafer, M. M., and Bergin, M.: Neither dust nor black carbon causing apparent albedo decline in Greenland's dry snow zone: Implications for MODIS C5 surface reflectance, Geophys Res Lett, 42, 9319-9327, https://doi.org/10.1002/2015gl065912, 2015.

This reference provides detailed characterization of Greenland dust properties, including the MAC values and hematite content we referenced. We will incorporate this citation in the final version.

Please let us know if any additional modifications would be helpful. We greatly appreciate your editorial oversight and the reviewers' insights that have strengthened our paper. Best regards,

Best regards,

Tenglong Shi and co-authors